# RDTS-Based Two-Dimensional Temperature Monitoring with High Positioning Accuracy Using Grid Distribution

**DOI:** 10.3390/s19224993

**Published:** 2019-11-16

**Authors:** Can Zhang, ZhongXie Jin

**Affiliations:** The Key Laboratory of Optoelectronic Technology & System, Education Ministry of China, Chongqing 400044, China; 20126981@cqu.edu.cn

**Keywords:** distributed temperature sensing, two-dimensional positioning, positioning accuracy

## Abstract

A novel two-dimensional (2D) positioning method based on Raman distributed temperature sensing (RDTS) has been reported to dramatically improve positioning accuracy. Using a well-designed 2D distribution of optical fiber and corresponding algorithms, the heat source can be accurately located without crosstalk; however, there is a tradeoff between sensing distance and positioning accuracy. In our experiments, an RDTS system with a spatial resolution of 0.8 m along a 3 km multimode fiber (MMF) is used with specific 2D routing rules and corresponding algorithms. A positioning accuracy of about 0.1 m is obtained without hardware modification, which could be improved through the dense arrangement of fiber; however, this would sacrifice the sensing length. This solution can be used for both flat surfaces and curved surfaces such as pipes or tank surfaces. This scheme can also be extended to three-dimensional positioning using a delicate routing design of sensing fiber.

## 1. Introduction

Raman distributed temperature sensors (RDTSs) based on spontaneous Raman backscattering (RBS) and optical time domain reflectometry (OTDR) have been widely used in pipeline monitoring, oil well monitoring, and fire alarm systems due to their outstanding ability to achieve temperature information with high resolution, high repeatability, and anti-electromagnetic interference [1,2,3,4,5,6]. In an RDTS, an interrogation pulse from the laser is injected into the fiber and the RBS light is used to obtain the temperature profile along the sensing fiber link (SFL) [7]. However, due to the weak intensity of RBS light, the RDTS usually obtains a Raman intensity about 70 dB weaker than the intensity of an injected laser, resulting in low spatial and temperature resolutions [8].

Over the past 30 years, to achieve long-distance and high-accuracy measurements of RDTS systems, several advanced techniques have been introduced, including optical pulse coding techniques [9,10], Rayleigh noise cancellation technology [11], and a multistage constant temperature controller [12]. Optical pulse coding techniques improve the signal-to-noise ratio (SNR) of RDTS systems, resulting in a nearly 10-fold resolution improvement. The Rayleigh noise cancellation technology improves temperature measurement accuracy by eliminating the Rayleigh noise in Stokes (S) and anti-Stokes (AS) light. The constant temperature control system makes a nearly two-fold temperature resolution improvement through the use of multistage thermostats and dynamic gain calibration technology. The above methods improved some kind of performance of RDTS, but only one-dimensional temperature information was obtained.

In recent years, 2D or 3D positioning using DTS has become increasingly necessary for commercial or military applications. Shellee et al. demonstrated a high-accuracy 2D RDTS using superconducting nanowire, single-photon detectors and single-photon counting techniques, enabling the best spatial resolution on the order of 1 cm [13]. Lomperski et al. introduced an optic sensing map that made the fiber sensor wind back and forth across the tank midplane to form the horizontal measurement section [14]. The fiber optic sensor readings were combined with particle image velocimetry (PIV) and infrared measurements to relate flow field characteristics to the thermal signature of the tank lid. Sun et al. installed part of two parallel fibers on the ceiling to locate the fire source in a room, which transferred 3D positioning to 2D plane [15]. Zhang et al. utilized the longitudinal lining technology of 3D temperature display to provide a solution for temperature field monitoring as well as fire visual localization of the tunnel through RDTS systems [16].

A traditional S-type 2D routing rule was used for plane positioning in the above multi-dimensional positioning methods. There was positioning accuracy improvement along the fiber laying direction, because positioning accuracy was determined by the laying interval. However, in the other direction, the positioning accuracy was limited by the original spatial resolution of RDTS. Since the S-type 2D positioning considered only horizontal or vertical signals, it missed the combination of both directions. Another limitation was the increasing cost of the system, due to the use of advanced materials, complemental hardware, or complex algorithms to improve the systematic spatial resolution.

In this paper, we propose an optimized, two-dimensional positioning method based on a traditional RDTS system to enhance the positioning accuracy. A 45°-inclined-cross-type (45°-ICT) routing rule and supporting algorithm are adopted to monitor two-dimensional temperature information. Modification in the algorithm instead of material or structure will reduce the costs of the RDTS system.

## 2. Principle of Two-Dimensional Positioning Based on RDTS

In RDTS, a high-power optical interrogation pulse generated by a laser source is injected into the sensing fiber link (SFL) and scattered. The RBS lights are divided into Stokes and anti-Stokes light by a wavelength division multiplexer (WDM) and are then detected by two avalanche photodiodes (APDs). The temperature information can be extracted from the classical equation [17]
(1)$$R\left(z\right)=\frac{I_{AS}\left(z\right)}{I_{S}\left(z\right)}=e^{-\left(\alpha_{AS}-\alpha_{S}\right)z}{\left(\frac{\lambda_{AS}}{\lambda_{S}}\right)}^{4}\exp\left[-\frac{h\Delta \nu }{k_{B}T\left(z\right)}\right]$$
where *R*(*z*) is the ratio coefficient between the intensity of AS (*I_AS_*(*z*)) and S (*I_S_*(*z*)) light; *α_AS_* (*α_S_*) and *λ_AS_* (*λ_S_*) are the absorption coefficient and wavelength of AS (S) light, respectively; Δ*ν* is the Raman frequency shift; *h* is the Planck constant; *k_B_* is the Boltzmann constant; and *T*(*z*) is the absolute temperature. It should be noted that the power of RBS light is usually 60–70 dB weaker than the peak power of injected pulse, which leads to a poor signal-to-noise ratio. As a result, a long averaging time is usually required for the RDTS to achieve a desirable temperature and spatial resolution.

The temperature information of one-dimensional RDTS can be easily calculated using the above formula. However, traditional two-dimensional distribution cases such as the S-type method, shown in Figure 1a, would not absolutely achieve a high-accuracy 2D positioning because there are no meaningful parameters better than the systematic spatial resolution in the longitudinal direction. Moreover, when the center of the heat source is located at the corner, as shown in Figure 1a (red star), it can be hard to distinguish between these two locations because of the similar temperature distribution of the SFL. To solve this problem, a novel 2D positioning principle was introduced and is described below.

A sufficiently long SFL was cross-distributed in 2D plane, dividing the 2D space into separate lattices, which were distributed as shown in Figure 1b. According to the 45°-inclined-cross-type rule, we kept the SFL route along the slope of 1 or −1, and the slope became the opposite when the SFL reached the edge of the sensing area. It was found that the intersecting fiber segments were far enough along the SFL, and the adjacent two intersections contained at least one pair of fiber segments that were far apart. Therefore, the distribution could significantly decrease the possibility of crosstalk. At the same time, the same positioning accuracy could be obtained in both directions.

While there were many ways to express the spatial position, a matrix representation method was proposed. As shown in Figure 1b, the plane was divided into many uniform lattices. The corners of the lattices (or the intersections of fiber) carried the temperature information of the SFL, which was used to define the spatial locations. The matrix corresponds to the intersections, the row and column values of the matrix correspond to the 2D coordinates of the intersections, and the values of the elements in the matrix are the temperature values of the intersections. The matrix was solved as follows.

The solution process of the matrix was realized by writing a piece of code. The core program sketch in a commercial mathematical software called MATLAB is shown in Figure 2. The DTS continuous output temperature vs. distance information, and the distance information combined with the starting point and lattice size, was used to establish the matrix. For the 45°-ICT pattern, the side length of each basic unit was *a* but the minimum fiber fragments containing temperature information depended on the systematic sampling frequency. In our system, a sampling frequency of 250 MHz corresponded to a temperature value per 0.4 m, which was mismatched with the side length in the pattern. Therefore, a linear interpolation method was necessary to estimate the temperature value of each fiber fragment. The original function between temperature and distance *F*(*z*) was translated into *G*(*z*), and then the refactored unary function was transferred into a binary function according the 45°-ICT pattern. It was found that the horizontal and vertical coordinate values of the space varied periodically as the pulse light traveled in the fiber, and the period of the coordinate was twice the length or width of the rectangular monitoring area. It is worth noting that the coordinate values varied in different half cycles (one increased, the other decreased). The surplus operator (*z*% *p*) was the period in horizontal (*p*_1_) and vertical (*p*_2_) used to resolve the coordinates. In general, there were two temperature values at per intersection, a higher temperature and a lower temperature. The numerical optimization method was to assign the lower temperature values to the intersections. This meant that because the lower temperature was high enough, the point was a high-temperature abnormal point. This method could effectively avoid the misjudgment of adjacent multiple points as high temperature, due to the limitation of system spatial resolution. *G*(*x*, *y*) was obtained using this method, as shown on the rightmost side of Figure 2. However, there were some zero elements in the matrix corresponding to the center points of the lattices that were replaced by the averaged values of the surrounding temperature to obtain a good display. In this way, the SFL with distance and temperature information regularly covered the entire sensing area.

## 3. System Configuration

The experimental setup is schematically shown in Figure 3. During the experiment, a high-power pulse laser source operating at about 1550 nm wavelength with a pulse width of 7 ns and a repetition rate of 8 kHz was used. The pulsed laser was launched into the SFL through a WDM, and then the weak backscattered lights along the SFL were detected simultaneously by two high-sensitivity, low-noise APDs with a bandwidth of 150 MHz. A data acquisition (DAQ) card with 12 digits and 250 MHz sampling frequency was used for signal processing, and the processed results were transmitted to personal computer (PC) for graphical display. In order to achieve a temperature precision of 1 °C 150,000 averaging times were necessary. The practical system spatial resolution was about 0.8 m, and the corresponding positioning accuracy was about 1.6 m. Based on a 3 km standard MMF, the SFL constructed the 2D sensing pattern as shown in Figure 3c. The fiber end was bent to a small radius to greatly attenuate light intensity and avoid reflection effect, and the SFL was distributed by a specific wiring principle for positioning of abnormal temperature in the rectangular area of 5.2 m × 1.6 m. Incandescent lamps, radiant heaters, and hot water bags were used as heat sources to simulate abnormal temperature fields of different sizes.

## 4. Results and Discussions

### 4.1. Matrix-Based Represention

In our experiment, multiple tests were accomplished and different positions were tested to certify and verify the accuracy and validity of the model. The positioning accuracy of this wiring pattern can be defined as half of the diagonal of the lattices (d/2). An auxiliary grid line with black and red colors was used to display the actual layout path of the SFL and its real-time temperature information, as shown in Figure 4. A threshold temperature value was set to distinguish the environmental and abnormal temperatures, which were expressed in black and red colors, respectively. After comparing different testing results, some typical cases were chosen to cover the majority of possible situations, including detections in edges, near the edges, and in corner positions. As shown in Figure 4a,c, inflection points in the edges or at the corners were treated as intersections. In Figure 4b, there were inflections and intersection points with high temperatures when the heat source was in the vicinity of the edge. It was judged that the real heat source was near the intersection point after setting a higher priority of intersections than inflections. A case of two closed heat sources was also tested, as shown in Figure 4d. It should be noted that it was hard to distinguish between the two heat sources, because four intersections existed with abnormal temperatures. Additional priority was set in the matrix according to the locations of intersections in the abnormal temperature fiber segments (near the center of high temperature fiber segments, a higher priority of location was set than near locations on the side). As a result, multiple heat sources were detected by the 45°-ICT distribution and the corresponding representation method. No crosstalk occurred under the scheme during several months of experiments.

### 4.2. Dense Distribution

In the above experiments, a positioning accuracy significantly lower than the original spatial resolution was obtained in 2D application. To verify the relationship between the positioning accuracy and the wiring density, a double-dense, distribution-based experiment was accomplished, as shown in Figure 5a,b. The same rectangle area as shown in Figure 1 was reconstructed, and the diagonal length of each lattice was reduced to 0.2 m. The three added fragments tagged as a_1_, a_2_, and a_3_ in Figure 5a were necessary to ensure the 45°-ICT distribution, and a 3D surface map was added to show a better profile of the temperature information. A positioning accuracy of about 0.1 m was obtained in this case, and the temperature and positioning errors are discussed below.

### 4.3. Error Analysis

The temperature resolution was not affected by the positioning methods because the temperature values in 2D plane were originally extracted from the DTS, and it was mainly determined by the distance of sensing fiber, the property of APDs, and noise reduction skills such as averaging times and wavelet techniques. However, the temperature accuracy was affected in some cases, because the size of the heat source must be greater than the original spatial resolution of the DTS (0.8 m). In pursuit of higher positioning accuracy, smaller heat sources were used in the experiments, leading to a lower measurement temperature. Therefore, the threshold value mentioned above should be flexibly set according to the actual situations. In addition, the use of interpolation would also slightly pull down the peak temperature, as shown in Figure 6a. Both linear and Gaussian interpolation methods were used, but there was no significant difference between the two interpolation methods.

Due to the relationship between higher density and better positioning resolution, the positioning accuracy could still be improved using the above methods. However, there were several factors that affected the accuracy. First, the drawing tool did not accurately match the geometric length of the actual routing, as shown in Figure 6b. Secondly, the length of the fiber did not correctly represent the actual geometric length. In consideration of the above impacts, the positioning accuracy (*ρ*_spatial_) of our experiments could be expressed as
(2)$$\rho_{\mathrm{spatial}}=\frac{\sqrt{2}}{2}\left[a+\left|\frac{z_{0}-z_{1}}{z_{0}}\right|\cdot a\right]$$
where *a* is the side length of the minimum cell, *z*_0_ is the actual total fiber length in sensing area, and *z*_1_ is the theoretical total length. In our experiments, *ρ*_spatial_ was 0.2058 and 0.1028 m under two different densities, respectively.

In order to verify the above theoretical positioning accuracy, a heat source with the same temperature was placed randomly in the smallest lattice, as shown in Figure 6c. The spatial positioning accuracy was the biggest error between the actual heat source location and the calculated location. Four colors were used to represent and distinguish the actual temperature center. Green points represented the centers of actual heat sources that were located by the 2D positioning method at the intersection below, and the red, blue, and brown points corresponded to the right, above, and left intersections, respectively. The experimental results are listed in Table 1 and show errors ranging from 0–0.211 m, so the actual spatial positioning accuracy was 0.211 m, which is consistent with the theoretical values. In the dense distribution case, the actual positioning accuracy was also consistent with the theoretical accuracy under the same positioning method.

## 5. Conclusions

In conclusion, a novel two-dimensional positioning method based on the RDTS system has been demonstrated with highly enhanced positioning accuracy. By a 45°-inclined-cross-type sensing fiber distribution and two supporting methods, a 0.1 m positioning accuracy was obtained by using a 0.8 m spatial resolution DTS. Theoretically, the positioning accuracy of the two-dimensional positioning method could be enhanced by a denser distribution of sensing fiber. This technology can be applied to the monitoring of pipeline leaks, leak-before-break detection, structural health monitoring, etc.

## Figures and Tables

**Figure 1 sensors-19-04993-f001:**
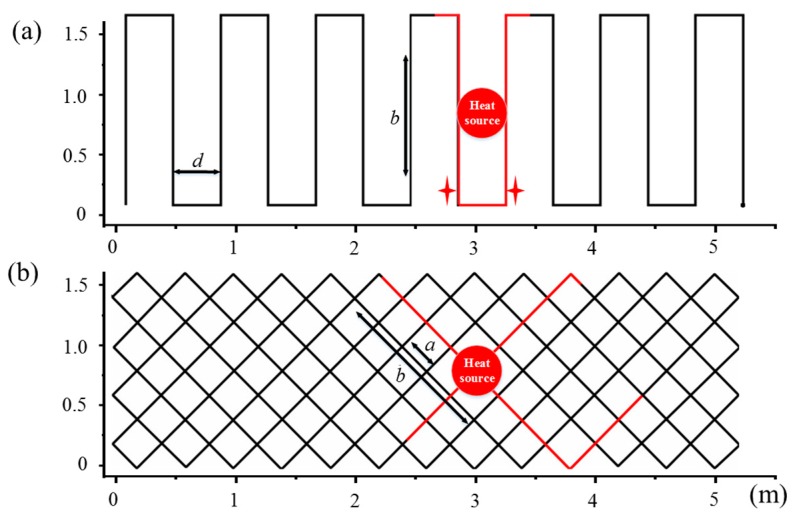
Two kinds of routing patterns: (**a**) S-type and (**b**) 45°-inclined-cross-type (45°-ICT). The parallel lines interval is presented by *d*, *b* is the spatial resolution, the red circular area and red star represent heat sources, and the minimum cell has a side length of *a* (*a* = *d*/$\sqrt{2}$). The red fiber segments correspond to the high temperature of the sensing fiber link (SFL).

**Figure 2 sensors-19-04993-f002:**
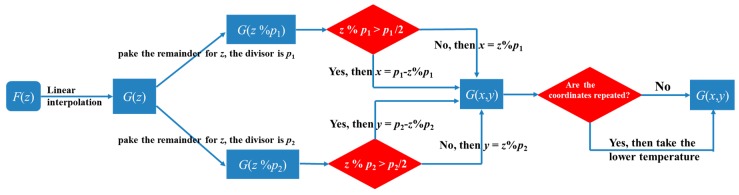
The core programming flow chart.

**Figure 3 sensors-19-04993-f003:**
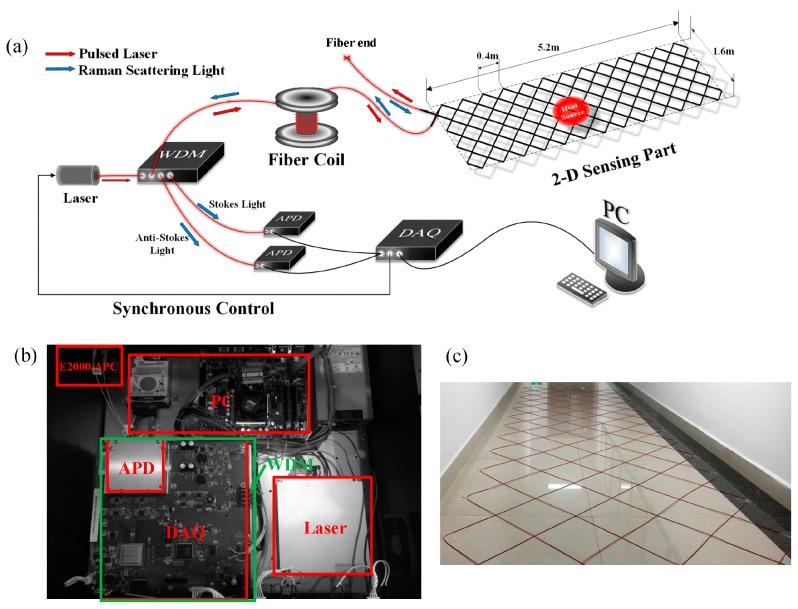
(**a**) Experimental setup of Raman distributed temperature sensing (RDTS) system. The 2D sensing part shows the specific wiring principle, wavelength division multiplex (WDM), avalanche photodiode (APD), data acquisition (DAQ), and personal computer (PC). The actual test platforms are shown in (**b**,**c**), and the green frame is the WDM under the DAQ.

**Figure 4 sensors-19-04993-f004:**
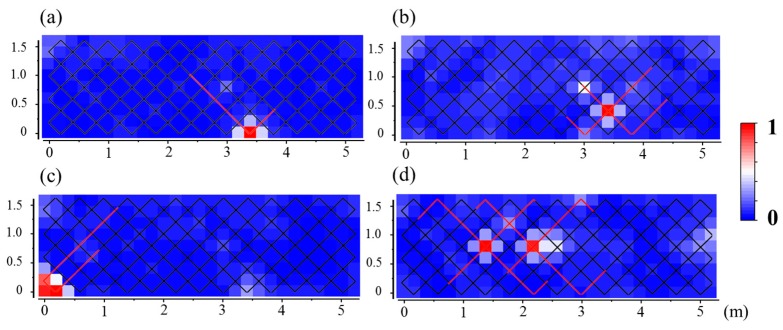
Different cases of heat source locations: (**a**) in the edge, (**b**) near the edge, (**c**) at the corner, and (**d**) a two-heat-sources case. An auxiliary grid line of the SFL was covered on the heat map.

**Figure 5 sensors-19-04993-f005:**
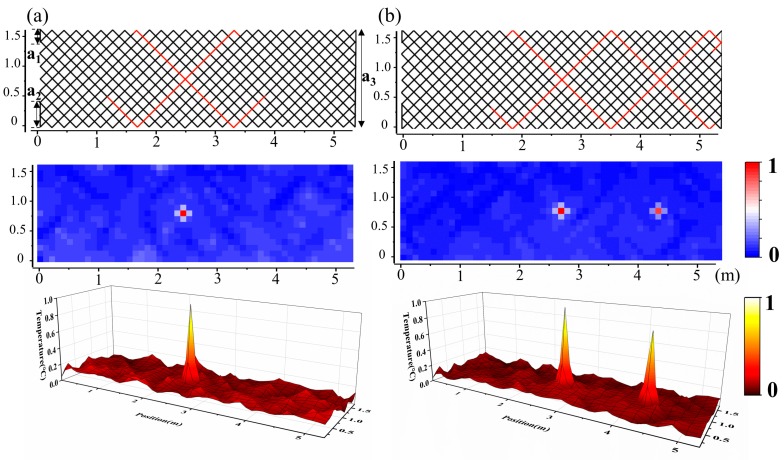
Temperature positioning under double-dense distribution: (**a**) one-source positioning and (**b**) two-sources positioning. Upper: auxiliary grid lines of SFL, middle: matrix representation method, and lower: 3D surface map.

**Figure 6 sensors-19-04993-f006:**
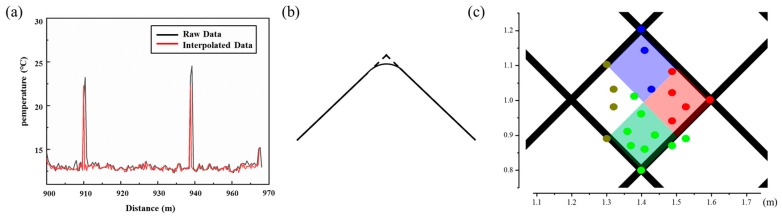
(**a**) Temperature versus distance curve: original data (black) and interpolated data (red); (**b**) difference between drawn graphics from theoretical (dotted line) and actual wiring curve (solid line); and (**c**) positioning accuracy test: green points represent the centers of actual heat sources by the 2D positioning method located at the intersection below, and the red, blue, and brown points correspond to the right, above, and left intersections, respectively.

**Table 1 sensors-19-04993-t001:** Error between the locations of actual heat source and calculated locations.

No.	Actual Loc. (cm)	Calculated Loc. (cm)	Error (cm)	No.	Actual Loc. (cm)	Calculated Loc. (cm)	Error (cm)
1	(153,89)	(140,80)	15.8	11	(149,94)	(140,80)	16.6
2	(136,91)	(140,80)	11.7	12	(153,98)	(160,100)	7.3
3	(132,98)	(120,100)	12.2	13	(160,100)	(160,100)	0
4	(137,87)	(140,80)	7.6	14	(149,102)	(160,100)	11.2
5	(140,80)	(140,80)	0	15	(143,103)	(140,120)	17.3
6	(141,86)	(140,80)	6.1	16	(149,103)	(160,100)	13.6
7	(130,89)	(120,100)	14.9	17	(141,114)	(140,120)	6.1
8	(140,96)	(140,80)	16	18	(140,120)	(140,120)	0
9	(149,87)	(140,80)	11.4	19	(138,101)	(140,80)	21.1
10	(144,90)	(140,80)	10.8	20	(132,103)	(120,100)	12.4

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
