# Peer review of "RDTS-Based Two-Dimensional Temperature Monitoring with High Positioning Accuracy Using Grid Distribution"

_sensors, 2019, doi:10.3390/s19224993_

Round 1
Reviewer 1 Report
The corrections have been provided by the authors. The paper can be accept.
Reviewer 2 Report
The revised work is suitable for publication.
This manuscript is a resubmission of an earlier submission. The following is a list of the peer review reports and author responses from that submission.
Round 1
Reviewer 1 Report
This paper introduces a novel abnormal temperature indoor positioning system. In general the paper has potential but it does not fulfill the requirements for a journal paper in the current status. In fact, the paper would be more suitable in a topic-related conference.
First, the introduction does not put the work in the appropriate context. In fact, 9 out of the 11 references cited in the paper are used to justify two sentences. I would have expected here the problem introduction, how it has been dealt in the past by other research teams and present your novel contributions, differentiating them from other related works. Also a related work section is missing, showing the advances and methods proposed by other authors.
Second, the description of the method proposed might not be detailed enough. Can a research reproduce your results in his/her own facilities with the description provided? If not, the proposed system is not reproducible and will have less interest for the readers. This sentence "Another processing method was to use intersections that could be encoded by two fiber fragments. Matrix and interpolation methods were used to increase the spatial resolution and the positioning accuracy. Both two methods will be analyzed in the fourth section." indicates that some details are missing. The description is vague and redirects the reader to section 4 "Results and Discussion" to understand how some methods were applied. I strongly recommend to have a comprehensive section describing the full method without any ambiguities.
Third, the results show some examples of your system working. The results are impressive in terms of accuracy but a comparison with other approaches has not been included. Moreover, the results are vaguely introduced. Sentences like "As shown in Figure 4(a), (b) and (c), the advanced method could also guarantee positioning accuracy. In Figure 4(d), this method could obviously avoid the false positioning in the two heat sources case." and "A positioning accuracy of about 0.1m has been obtained using 0.8m spatial resolution DTS" is not precise enough and only reflects examples on the evaluation. I would have expected a more comprehensive evaluation.
As I said, the work is promising but probably Sensors is not its forum.
Reviewer 2 Report
The paper proposed an optimized two-dimensional positioning method (45°-inclined-cross type) using Raman DTS system to enhance the positioning accuracy.
The authors commented in the introduction section: “Few relative researches on low-cost 2D positioning technique based on 41 distributed sensing were reported”. In fact there is a lack of papers of 2D Raman DTS in the literature review of the manuscript. The authors should cite and comment other works of 2D Raman DTS published in the literature, such as:
- Raman Distributed Temperature Sensor with Optical Dynamic Difference Compensation and Visual Localization Technology for Tunnel Fire Detection - DOI: 10.3390/s19102320. Please consider also the reference section of this paper.
- Fiber optic distributed temperature sensor mapping of a jet‑mixing flow field – DOI:10.1007/s00348-015-1918-6.
In the sections 4.1, 4.2 and 4.3 is not much clear how the matrices are constructed. The authors should write mathematically the formulation of these matrices in order to clarify this issue. In addition the authors should create a specific section for the methodology and other section for results. In the current manuscript both methodology and results are mixed.
The authors commented in the section 4.4 that the “The temperature resolution would not be affected by the positioning methods”, however it is known that RDTS systems has the resolution dependent on the distance. How this effect could affect the resolution in different parts of the proposed 2D mesh?
Reviewer 3 Report
This work presents a 2D positioning system for the detection of abnormal temperature by means of Raman distributed temperature sensing (RDTS) system. The paper lacks novelty and a clear comparison with existing literature.
The following issues need to be carefully addressed by the authors:
I suggest concluding the paper title with one of the following words “sensing/monitoring/detection”. The introduction with the state of the art should be enriched, the number of references reported is quite limited. Lines 72-79 are not clear. The methods reported should be clearly described, as for example “interpolation method”, “numerical optimization method” (lines 135-136), “drawing tool” (line 184), etc. The novelty of the proposed method is not clear and must be properly illustrated. There is no numeric comparison with the known state of art.Minor changes:
Line 121, I think it is Figure 3(a).